# Our Experience on Temporal Bone Fractures: Retrospective Analysis of 141 Cases

**DOI:** 10.3390/jcm10020201

**Published:** 2021-01-08

**Authors:** Filippo Ricciardiello, Salvatore Mazzone, Giuseppe Longo, Giuseppe Russo, Enrico Piccirillo, Giuliano Sequino, Michele Cavaliere, Nunzio Accardo, Flavia Oliva, Pasquale Salomone, Marco Perrella, Fabio Zeccolini, Domenico Romano, Flavia Di Maro, Pasquale Viola, Rosario Cifali, Francesco Muto, Jacopo Galli

**Affiliations:** 1Ear Nose Throt Departement AORN Cardarelli, 80100 Napoli, Italy; filipporicciardiello@virgilio.it (F.R.); ariete_gr@libero.it (G.R.); giuseq@gmail.com (G.S.); nunzioaccardo@libero.it (N.A.); flavia.oliva@arubapec.it (F.O.); pamka2@hotmail.it (P.S.); domenicoromanorl@gmail.com (D.R.); 2General Menager AORN Cardarelli, 80100 Napoli, Italy; direzione.generale@aocardarelli.it; 3Department of Otology & Skull Base Surgery, Gruppo Otologico, 29121 Piacenza, Italy; enrico.piccirillo@gruppootologico.it; 4Unit of Otorhinolaryngology, Department of Neuroscience, Federico II University Hospital, 80138 Napoli, Italy; miccaval@libero.it; 5Anesthesiology and Reanimation Department AORN Cardarelli, 80100 Napoli, Italy; marco.perrel@gmail.com (M.P.); rosariocifali@libero.it (R.C.); 6Radiology Department AORN Cardarelli, 80100 Napoli, Italy; fabio.zeccolini@aocardarelli.it (F.Z.); francesco@muto.it (F.M.); 7Otolaryngology-Head and Neck Surgery Department, University Hospital of Verona, 37132 Verona, Italy; fldm22@gmail.com; 8Unit of Audiology, Department of Experimental and Clinical Medicine, Magna Graecia University, 88100 Catanzaro, Italy; pasqualeviol@libero.it; 9Institute of Otolaryngology, Head and Neck Surgery, School of Medicine and Surgery, Università Cattolica del Sacro Cuore, 00168 Rome, Italy; jacopo.galli@iol.it

**Keywords:** temporal bone fractures, skull base, otosurgery, facial nerve paralysis, cerebrospinal fluid leakage

## Abstract

Temporal bone fractures are a common lesion of the base of the skull. The diagnosis and management of temporal bone fractures require a multidisciplinary approach. Variable clinical presentations may arise from such fractures, ranging from an asymptomatic course to very serious consequences. The aim of this study was to report our experience with a series of patients with temporal bone fractures and to propose a diagnostic/therapeutic algorithm. This study enrolled 141 patients, 96 (68.1%) males and 45 (31.9%) females, ranging in age from 20 to 60 (average age: 39 ± 4.1 years), with temporal bone fractures who were referred to Cardarelli Hospital between 2006 and 2018. The present paper presents a classification of temporal bone fractures and typical clinical sequelae and provides an illustration of their prognosis and treatment.

## 1. Introduction

Temporal bone fractures (TBFs) represent 14–22% of cranial fractures [1,2,3,4,5,6,7,8]. In the majority of patients, these fractures are unilateral, but approximately 12% of TBFs are bilateral [8,9]. TBFs have been extensively studied because of their severity and emergent nature [1,8,10]. Conventionally, TBFs are divided into transverse, longitudinal, and mixed, depending on the direction of the fracture line compared to the long axis of the petrous pyramid [1,8,9,10,11]. Some authors have criticized this classification because many fractures do not strictly follow this scheme [8,9,10,11,12,13]. An alternative classification system grouped TBFs as those sparing the otic capsule (OSC) or violating the otic capsule (OVC), because its involvement is a source of severe complications, which negatively affect patient quality of life (QoL) [1,8,9,10,12,13,14]. Approximately 95% of all TBFs are OSC fractures [1,10,12]. The diagnosis and management of TBFs must be made in a short time, as they deeply influence the prognosis. The close collaboration between anesthetists, neurosurgeons, radiologists, and otolaryngologists is of fundamental importance [1,8].

The aim of this study was to report our experience on TBF treatment.

## 2. Experimental Section

In this retrospective study, we consecutively enrolled 141 patients with TBFs, 96 (68.1%) males and 45 (31.9%) females, ranging in age from 20 to 60 (average age: 39 ± 4.1 years), who were referred to our hospital between 2006 and 2018.

We considered two stages:Initial stage: neurosurgeons and intensivists work to stabilize vital signs and assess the neurological status of the patient;Secondary stage: neurologists and otolaryngologists work to perform an otoneurological evaluation to treat any neurological outcomes and otological complications.

We classified TBFs as translabyrinthine TBFs (TL-TBFs, corresponding to OVC fractures) and extralabyrinthine TBFs (EL-TBFs, corresponding to OSC fractures). We considered bony labyrinth (otic capsule) involvement as the only element of differentiation.

After the initial stage of achieving stable vital functions, all the patients underwent the following examinations:high-resolution computed tomography (HRCT) examination of the temporal bone;otological examination;pure tone audiometry (PTA);vestibular examination;electromyography (EMG).

The recorded data included the following: mode of trauma, patient age, mortality, fracture pattern (TL-TBF or EL-TBF), degree of temporal bone pneumatization (none, mild, or complete), and otological features and their treatment. The degree of temporal bone pneumatization was classified into three groups with the ascending carotid artery on the axial plane used as the reference structure: no pneumatization (no evidence of pneumatization in the petrous apex), mild pneumatization (mild pneumatization either medial or lateral to the carotid canal), and complete pneumatization (complete pneumatization surrounding the carotid canal) [15].

PTA, performed as soon as the clinical conditions allowed it, was used to characterize the type of hearing loss (transmissive, sensorineural, or mixed) and quantify its extent (mild, medium, severe, and profound/anacusis) depending on the arithmetic mean of the air conduction threshold calculated at frequencies of 500, 1000, 2000, and 4000 Hz. PTA was measured using a Piano Clinical Audiometer (Inventis, Padua, Italy).

Vestibular examination included eye movement evaluation with and without fixation by videonystagmoscopy with video-oculography goggles (GN Otometrics, Taastrup, Denmark). The patient was evaluated in the sitting position and in six different positions according to the standardized procedure (upright, gaze straight ahead, supine position with head elevation 30° nose up, supine position with head elevated 30° and head turned 45° left, supine position with head elevated 30° and head turned 45° right, backwards head in the hanging position turned 45° towards the right, backwards head in the hanging position turned 45° towards left.

EMG was used for the bilateral and comparative detection of spontaneous and voluntary action potentials or fibrillation potentials associated with stimulation/detection tests. Detection electrodiagnosis was performed with a needle electrode in the orbicularis oris and orbicularis oculi muscles. The tests were performed two weeks after facial nerve paralysis. In the case of neurotmesis, fibrillation potentials were found, and no response was obtained after stimulation. This pattern was typical of complete degeneration. In the case of neurapraxia, a blocked conduction pattern was caused by a demyelination process: no voluntary action potential was found, but through nerve stimulation, a synchronized evoked response was obtained.

Non-otoneurologic symptoms were excluded from the study.

MedCalc Statistical Software version 19.1.7 (MedCalc Software bvba, Ostend, Belgium; https://www.medcalc.org) was used to perform statistical analysis. Our data were tested with the Student’s *t*-test. *p* < 0.05 was considered statistically significant.

## 3. Results

Of the 141 patients enrolled in the study, 13 died (9.2%) (11—during the initial stage; 2during the secondary stage).

The examined sample included 128 patients, 87 (68%) males and 41 (32%) females, ranging in age from 20 to 60 (average age; 35 ± 4.1 years).

The trauma causes were road accidents, 72 (56%); accidental falls (also work accidents), 27 (21%); assaults, 23 (18%); and gunshot-related injuries, 6 (5%).

HRCT revealed 105 cases of EL-TBF (82%) and 23 cases of TL-TBF (18%) (Figure 1). Bilateral fractures occurred in 4 patients (3.1%) and all were EL-TBFs.

The degree of temporal bone pneumatization in patients with TL-TBFs and EL-TBFs is shown in Table 1.

In the examined sample, we detected the following signs and symptoms (Table 2):otorrhagia—in 115 cases (89.84%);hemotympanum—in 100 cases (78.12%);vertigo—in 86 cases (67.18%);conductive hearing loss (CHL)—in 70 cases (54.68%);sensorineural hearing loss (SNHL)—in 30 cases (23.43%);perforation of the tympanic membrane (TM)—in 45 cases (35.15%);facial nerve paralysis (FNP)—in 11 cases (8.59%);cerebrospinal fluid (CSF) leakage—in 9 cases (7.03%).

The otorrhagia in all the patients was treated with antibiotics, otomicroscopy-guided blood aspiration, and external acoustic meatus medication.

The hemotympanum required only clinical observation; it spontaneously resolved in 3–4 weeks (100% of cases).

Vertigo occurred in 86 patients: 64 (74.42%) had canalolithiasis and 22 (25.58%) had acute vestibular deficit (AVD). Canalolithiasis was treated with a canaltih repositioning procedure maneuver and AVD was treated with pharmacotherapy and vestibular rehabilitation. Among the patients who suffered canalolithiasis, dizziness persisted for more than three months in 38 (59.37%). They were treated with vestibular rehabilitation and the symptoms resolved in all cases within 18 months. In patients with AVD, vestibular compensation occurred within 4–7 months.

A total of 100 patients (78.1%) had hearing loss.

CHL occurred in 70 patients (69 EL-TBF + 1 TL-TBF) and spontaneously resolved, except for 45 patients with TM perforation; 30 (66.7%) patients experienced spontaneous resolution within a month; myringoplasty was recommended for 15 patients (33.3%) four months after the fracture; only 3/15 patients accepted surgery (20%).

SNHL occurred in 30 patients and was irreversible in all cases.

Regarding the hearing loss classification, depending on the severity (mild, medium, severe, or anacusis), in EL-TBF patients, we observed 20 cases of mild hypoacusis, 42 cases of medium hypoacusis, and 16 cases of severe hypoacusis, while in TL-TBF patients, we observed 3 cases of severe hypoacusis and 19 cases of anacusis (Table 3).

A total of 11 patients (8.59%) presented with FNP. Four (36.36%) patients had immediate FNP; the House–Brackmann score (HB-S) ranged from 5 to 6.

Six (54.54%) patients had delayed FNP (3 patients had an HB-S of 3, and 3 patients had an HB-S of 5). In 1 (9.10%) patient with FNP with an HB-S of 5, we had no information on the time of onset. A two-week corticosteroid treatment regimen was initiated in all the patients who presented with FNP; in all the patients with delayed paralysis and in the patient with paralysis of an unknown onset, there was complete resolution within 3–6 months; in all our cases, EMG was performed after two weeks and then every month, EMG data excluded neurotmesis. Only one patient with immediate paralysis experienced resolution with corticosteroid treatment after two months. In the remaining three cases with persistent paralysis, showing typical EMG patterns of nerve degeneration, surgical therapy was recommended due to the lack of response to corticosteroid treatment. These three patients had profound SNHL, we utilized a translabyrinthine approach with facial nerve decompression from the stylomastoid foramen to the geniculate ganglion. In these patients, the postoperative functional recovery was partial (recovery from an HB-S of 5–6 to an HB-S of 3).

A total of 9 patients had CSF leakage; all of them had a TL-TBF. These patients were treated with antibiotics and conservative measures, such as a compressive head bandage for at least a month; resolution occurred within 1–2 weeks in 7 cases (77.78%). Subtotal petrosectomy with obliteration by autologous fat was necessary for two patients with persistent vertigo, CSF leakage, and risk of meningitis with complete resolution after surgery.

## 4. Discussion

Head trauma continues to be a major cause of morbidity and mortality. TBFs carry a relatively high risk of mortality, ranging from 10–27%, predominantly due to associated intracranial injuries [5,9,11,16,17].

TBFs are conventionally classified as longitudinal or transverse according to the relationship of the fracture line and the long axis of the petrous pyramid; fractures that course parallel are considered longitudinal, and fractures that course perpendicular are considered transverse [2,8,10].

A more clinically relevant classification describes fractures relative to otic capsule involvement. Fractures are either OSC or OVC [1,10,12,13]. The otic capsule corresponds to labyrinthine bone, so the authors refer to the same classification but with a different denomination (translabyrinthine instead of OVC and extralabyrinthine instead of OSC). These denominations were used to specify the site of FNP in TBFs as translabyrinthine (perigeniculate ganglion area) and extralabyrinthine (tympanic and mastoid tracts) [18,19,20]. However, this classification has a weak point in the prevalence of EL-TBFs in our study (82% of our sample’s TBFs).

In all classifications, HRCT of the temporal bone represents the gold standard for diagnosis and classification. Our data also demonstrate that TL-TBFs are predominantly observed with mastoid hypopneumatization (Table 1). According to the literature, the authors endorse a potential protective effect of pneumatization in TBFs; in fact, pneumatic cells can absorb most of the impact force during a traumatic event [21,22]. CHL is frequently observed in TBF cases; it is often secondary to otorrhagia, hemotympanum, TM perforation, or ossicular chain disruption [1,2,9,10,23]. Our data show a significant association between CHL and EL-TBFs. In our study, the management of the causes of CHL did not present remarkable difficulties. Otorrhagia and hemotympanum evolved towards resolution. The tympanic perforation spontaneously resolved in most cases. HRCT does not always allow the diagnosis of ossicular discontinuity; therefore, persistent CHL with an intact TM is an indication for surgical exploration, eventually followed by ossiculoplasty [9]. In our study, three patients had a suspected ossicular discontinuity, but they did not accept surgical treatment. SNHL is more frequent in TL-TBFs and is irreversible [1,2,8]. Our data demonstrate that a significantly greater proportion of the patients with TL-TBFs (95.7%) had SNHL than among the patients with EL-TBFs (7.6%). Our data confirm that SNHL is often severe in TBFs (anacusis or severe hypoacusis in 22/23 cases).

Vertigo frequently occurs after temporal bone trauma and its incidence is underestimated due to the overlap with neurological symptoms [7,8,9,10,23]. It may be secondary to AVD due to vestibular fractures in TL-TBFs or to post-traumatic otolithic detachment in EL-TBFs. Dizziness can persist for several weeks due to delayed vestibular compensation in AVD or to utricular damage in patients with an intact labyrinth. In TL-TBF patients with AVD, vestibular compensation occurred in most patients within 3–6 months [24,25].

Our data show that vertigo from canalolithiasis was the most frequent.

CSF leakage is due to fracture of the medium fossa and/or otic capsule. Its incidence is underestimated owing to concomitant otorrhagia, which makes it difficult to recognize at an early stage. The literature reports that the incidence of CSF leakage in TBF patients ranges from 11% to 45% [1,2,8,10,13]. Most of these cases represent CSF otorrhea; in a few cases, an intact TM will shunt the fluid to the eustachian tube with resultant CSF rhinorrhea. If the diagnosis is less obvious, the differential diagnosis is based on the β-2 transferrin level, which is considerably high in CSF.

The treatment of CSF leakage begins with conservative measures, including head elevation (20–30°), bed rest, stool softeners, avoidance of nose blowing/sneezing, compressive head bandages, and, in selected patients, placement of a lumbar drain [1,8,9,10,26].

The use of prophylactic antibiotics to reduce the risk of meningitis is controversial [1,10,14,27]. A 2011 Cochrane review that examined evidence from 208 patients from 5 randomized controlled trials did not demonstrate any significant difference between the antibiotic prophylaxis and control groups [28]. We used antibiotic prophylaxis in all the patients with CSF leakage for two weeks. The literature reports that only a small percentage of patients require surgical repair; most CSF leakage cases resolve spontaneously within one week [1,8,10,26]. According to many authors, we believe that surgical treatment should be considered after the failure of conservative approaches [1,5,6,9,10,13,29]. Our data show that 9 patients with TL-TBFs had CSF leakage. Seven cases (77.8%) evolved towards resolution with conservative treatment within two weeks, and two patients underwent subtotal petrosectomy with obliteration of the middle ear and the eustachian tube.

Several studies have demonstrated that the incidence of FNP is estimated to be between 30% and 50% in TL-TBF and between 6% and 13% in EL-TBF [1,2,8,10,13,18]. In 2001, Darrouzet et al. [20] published a report on 115 cases of post-traumatic FNP (TBFs accounting for 34.3% of cases); most patients (77.7%) had complete FNP, and 59 cases (51.3%) had complete FNP immediately. We observed only 11 (14.7%) patients with FNP, and according to the literature, FNP is prevalent in patients with TL-TBFs. The difference in FNP prevalence between TL-TBFs and EL-TBFs is statistically significant. Within our series, 4 patients had immediate FNP, and 6 patients had delayed FNP; in 1 case, we had no information on the time of onset. Patients with TL-TBFs were significantly more likely to have immediate FNP than patients with EL-TBFs (*p* = 0.003). This difference is of fundamental importance in determining the prognosis: early paralysis generally does not culminate in spontaneous resolution since it is caused by an interruption of nervous conduction in a nerve section or nerve compression by bone segments; conversely, a tardive insurgence of FNP is often attributable to post-traumatic neural edema or hematoma, which in most cases results in spontaneous resolution [1,30]. In each case in which a tardive insurgence of paralysis was observed, we recorded spontaneous resolution with medical therapy within 3–6 months. In patients with early paralysis and EMG patterns of nerve degeneration, we considered whether to perform surgery. The timing of facial surgery is controversial and should be guided by clinical presentation, neuroradiological findings and electrodiagnostic tests. Several studies have argued against surgical decompression of delayed post-traumatic FNP because it is largely accepted that these patients will achieve a complete recovery [1,9,10]. The benefits of surgery performed within 3 or 8 months [23,31] are confirmed by several studies. We surgically treated three patients with complete and early FNP and documented lesions on CT after 2 months; in two patients, EMG showed nerve degeneration. Facial function remained poor in one of the operated patients. We endorse a conservative approach to treating facial nerve injuries and suggest surgery for patients in whom complete FNP persists for 6–8 months after trauma with patterns of nerve degeneration on EMG and for patients in whom a nerve section is shown on HRCT. The surgical approach is guided by the site of the lesion. It might be important to remember that approximately 80% of injuries to the facial nerve occur at the level of the geniculate ganglion; therefore, in most cases of severe hearing loss, a combined middle cranial fossa/transmastoid approach or translabyrinthine approach is adopted [1,10,26]. A recent study proposed a transcanal approach to surgical decompression of the facial nerve in the perigeniculate ganglion area, but we did not have any experience with this approach [32].

## 5. Conclusions

The treatment of TBFs is extremely complex and requires an appropriate clinical framework from the moment of the traumatic event to plan a multidisciplinary approach with perfect coordination among various specialists (otolaryngologists, neurosurgeons, neurologists, neuroradiologists). The treatment and management of associated life-threatening intracranial pathologies should be the primary focus. The management of otoneurological complications should be addressed once clinical stability is achieved. Meticulous knowledge of every possible complication that could follow the traumatic event is required to prevent them and obtain the best possible result.

Overall, the patient prognosis depends on intracranial involvement. Treatments for common TBF complications are as follows:FNP: Partial and/or delayed facial paralysis has a favorable overall prognosis with conservative treatment only. Early paralysis generally does not culminate in spontaneous resolution since it is caused by an interruption of nervous conduction for a nerve section or nerve compression by bone segments. The monitoring of neural function by EMG is crucial. The invasive character of the surgical approach used to treat geniculate ganglion lesions justifies the rigorous selection of patients who might benefit from surgery in terms of prognosis. This selection is based on clinical (time of onset), radiologic (fracture line), and EMG data. Indications for surgical exploration and decompression are as follows:
complete paralysis resulting from a nerve section documented on CT;complete paralysis with EMG-documented degeneration;complete facial paralysis that persists for 6–8 months after trauma with EMG signs of degeneration;CSF leakage: Most CSF leaks will resolve within two weeks with only conservative measures, including head elevation (20–30°), bed rest, stool softeners, avoidance of nose blowing/sneezing, compressive head bandages, and, in selected patients, placement of a lumbar drain. Surgical repair of the leak is possible in a few refractory cases.Hearing loss: Hearing loss is most commonly conductive in nature and its management has not presented remarkable difficulties. SNHL is less frequent and is irreversible.Vertigo: Vertigo often has an advantageous prognosis.

## Figures and Tables

**Figure 1 jcm-10-00201-f001:**
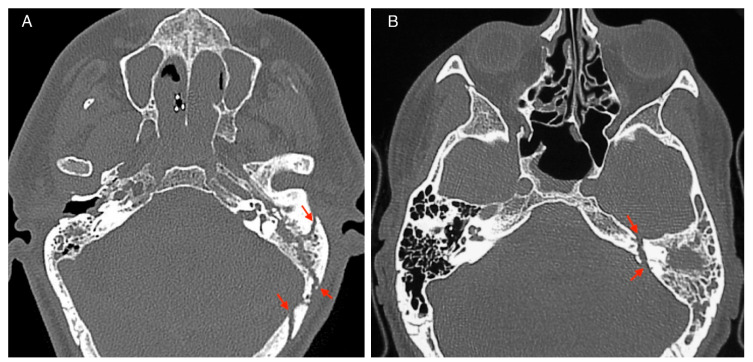
Temporal bone fracture’s HRCT imaging: (**A**) extralabyrinthine, (**B**) translabyrinthine Red arrows indicate the fracture’s line.

**Table 1 jcm-10-00201-t001:** Degree of temporal bone pneumatization in patients with a temporal bone fracture.

	TL-TBF	EL-TBF
None	16	2
Mild	8	64
Complete	0	38

**Table 2 jcm-10-00201-t002:** Classification of signs and symptoms in temporal bone fractures. Vertigo, otorrhagia, and hemotympanum are the most common signs in both EL and TF temporal bone fractures. Sensorineural hearing loss is the most common in TL-TBF, while conductive hearing loss is often found in patients with EL-TBF.

	EL-TBF (*n* = 105)	TL-TBF (*n* = 23)	All TBF (*n* = 128)	*p*-Value
Vertigo	64 (60.95%)	22 (95.65%)	86 (67.18%)	0.0013
Otorrhagia	94 (89.52%)	21 (91.30%)	115 (89.84%)	0.7979
Hemotympanum	82 (78.09%)	18 (78.26%)	100 (78.12%)	0.9861
Sensorineural hearing loss (HL)	8 (7.61%)	22 (95.65%)	30 (23.43%)	0.0001
Conductive HL	69 (65.71%)	1 (4.34%)	70 (54.68%)	0.0001
TM perforation	37 (35.23%)	8 (34.78%)	45 (35.15%)	0.9669
FNP	2 (1.90%)	9 (39.13%)	11 (8.59%)	0.0001
CSF leak	0 (0%)	9 (39.13%)	9 (7.03%)	0.0001

**Table 3 jcm-10-00201-t003:** Classification of hearing loss in patients with TBF: TL-TBF seems to be associated with a worse hearing loss, compared with EL-TBF. As shown in Table 2, CHL is mostly represented in EL-TBF, while TL-TBF is highly associated with SNHL. A PTA threshold of 20–40 dB is considered mild, 41–70 dB—moderate, 71–90 dB—severe, and a PTA > 91 dB—profound. Most of our patients with TL-TBF suffered of anacusis.

	TL-TBF	EL-TBF
Mild	0	20
Moderate	0	42
Severe	3	16
Profound/anacusis	19	0

## Data Availability

Data is contained within the article.

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
