# Peer review of "Our Experience on Temporal Bone Fractures: Retrospective Analysis of 141 Cases"

_jcm, 2021, doi:10.3390/jcm10020201_

Round 1

Reviewer 1 Report

Interesting work with a reasonable number of cases described in a retrospective narrative manuscript.

With 17 authors, what was the role of each author?

Retropective work with 141 cases after 12 years. Authors don´t explain if they are consecutive cases.

Several items in results are displayed two times, both in text and graphs.

Several graphs are not self explanatory.

We cannot see the new (?) "classification of temporal bone fractures" that seemed to be the purpose of the manuscript, with a consequent improvement from what is already know for decades. OSC or OVC are direct consequences of the frature with obvious implications in treatment and prognosis.

Author Response

  • With 17 authors, what was the role of each author?

Dear Reviewer, thanks for careful review of our manuscript. I attach the role of each author:

Filippo Ricciardiello and Salvatore Mazzone: Conceptualization, writing original draft preparation.

Pasquale Viola, Giuseppe Russo and Giulio Sequino: Conceptualization, writing reviewing and editing.

Michele Cavaliere, Enrico Piccirillo and Flavia Oliva: Software and visualization.

Pasquale Salomone, Marco Perrella and Fabio Zeccolini: Methodology and visualization.

Domenico Romano, Flavia Di Maro and Nunzio accardo: Software and validation.

Rosario Cifali,  Giuseppe Longo and Francesco Muto: Formal analysis and visualization.

  • Retropective work with 141 cases after 12 years. Authors don´t explain if they are consecutive cases.

Dear Reviewer, thanks for careful review of our manuscript. it is a retrospective study with 141 consecutive patients.

  • Several items in results are displayed two times, both in text and graphs. Several graphs are not self explanatory.

Thanks for this comment. We have deleted the graphs and improved the tables in order to make them clearer. Legends with full explanation of the tables have been included. We really appreciated your careful and thoughtful evaluation of our manuscript and hope that this revised version meets with your approval.

  • We cannot see the new (?) "classification of temporal bone fractures" that seemed to be the purpose of the manuscript, with a consequent improvement from what is already know for decades. OSC or OVC are direct consequences of the frature with obvious implications in treatment and prognosis.

We classified TBFs as translabyrinthine (TL-TBFs, corresponding to OVC fractures) and extralabyrinthine (EL-TBFs, corresponding to OSC fractures).  Our classification is not a new classification but reflects the subdivision into OVC and OSC using different names which are, in our opinion, easier to remember and memorize.These denominations were used by Fish to divide the facial nerve into extralabyrinthine and translabyrinthine based on the lesion site.

Reviewer 2 Report

Overall, this is a well-articulated manuscript. The authors have done a prolific job in collecting the data, classifying different types of temporal bone fractures, clinical outcome and treatment. However, the authors need to address the following:

Experimental section:

  1. In line 61-66, the authors mentioned various techniques used for this study. However, following that only few of those are detailed out. Please describe all the techniques/ parameters used for this study in detail for example delineate how or what kind of vestibular tests were conducted.
  2. A detail description of statistical analysis is warranted.

Result section 

  1. Figure 2 depicts the degree of pneumatization. Does the graph explain a linear trend for EL-TBF? Please explain.
  2. Similarly, in the figure 5 is there a linear trend for mild to severe hearing loss in the patients with EL-TBF injury?
  3. Towards the end of the result section all the results are not described in an organized fashion. In my opinion, it can be arranged in an orderly manner for example, mention all the ear injuries and hearing loss in one paragraph, otorrhagia and similar things in another paragraph, so on and so forth.
  4. Spelling correction for table 2 is required. 

Author Response

  • In line 61-66, the authors mentioned various techniques used for this study. However, following that only few of those are detailed out. Please describe all the techniques/ parameters used for this study in detail for example delineate how or what kind of vestibular tests were conducted.

Thanks for this comment. All the techniques/parameters used for this study were added in detail.

  • A detail description of statistical analysis is warranted.

Thanks for pointing this out. We have added a detail description of statistical analysis.

  • Figure 2 depicts the degree of pneumatization. Does the graph explain a linear trend for EL-TBF? Please explain. Similarly, in the figure 5 is there a linear trend for mild to severe hearing loss in the patients with EL-TBF injury?

Thanks for this comment. We have deleted the graphs and improved the tables in order to make them clearer. Legends with full explanation of the tables have been included.

  • Towards the end of the result section all the results are not described in an organized fashion. In my opinion, it can be arranged in an orderly manner for example, mention all the ear injuries and hearing loss in one paragraph, otorrhagia and similar things in another paragraph, so on and so forth.

Dear Reviewer, following your suggestions we have ordered the results section in order to make them clearer.

  • Spelling correction for table 2 is required.

Thanks for pointing this out. Spelling correction for table 2 has been made.

Reviewer 3 Report

The present study covers 141 temporal bone fracture patients from 2006 to 2018. Thirteen patients initially died and thus the outcome of 128 patients were followed throughout the study. The patients were grouped according to translabyrinthine fractures (affecting the otic capsule) and extralabyrinthine fractures (not affecting the otic capsule), 23 and 105 cases, respectively. The patients were well investigated regarding CT, degree of temporal bone pneumatization, otorrhagia, haemotympanum, vertigo, hearing loss – HL (conductive and sensorineural), tympanic membrane perforation, facial nerve paralysis (FNP) and cerebrospinal fluid (CSF) leakage. As expected the translabyrinthine fractures caused more serious outcomes; CSF leakage, permanent HL and permanent FNP.

My comments are the following:

  • Is this study a retrospective or prospective study? Any information concerning e.g. investigations of medical records cannot be found.
  • - Any information about when follow ups were performed regarding the various symptoms and signs cannot be found
  • - Regarding conductive hearing loss a follow up at a later stage to evaluate the final hearing outcome had been of interest.
  • - Most results are presented both as a table and a graph (Fig 2 and Table 1, Fig 4 and Table 2, Fig 5 and Table 3) or as Fig 3 and complete results text. Once is enough!
  • - The Discussion is very talkative and could be reduced in length. -
  • Is the title of the study relevant? Nowhere in the text I can find any clear remarks that this is really in the expertise field of the otorhinolaryngologist.

Author Response

  • Is this study a retrospective or prospective study? 

Dear Reviewer, thanks for careful review of our manuscript. it is a retrospective study with 141 consecutive patients .

  • Any information about when follow ups were performed regarding the various symptoms and signs cannot be found. Regarding conductive hearing loss a follow up at a later stage to evaluate the final hearing outcome had been of interest.

Thanks for pointing this out.

Haemotympanum spontaneously resolved in 3-4 weeks in all patients.

Canalolithiasis was treated with canaltih repositioning procedure maneuver with quite satisfactory results. Dizziness persisted for more than three months in 38 patients and were treated with vestibular rehabilitation with good results within 18 months. In patients with AVD, after therapy, vestibular compensation occurred within 4-7 months. 

Conductive hearing loss spontaneously resolved, except for in 45 patients with TM perforation and 3 patients with suspected ossicular discontinuity who refused surgical treatment. Hearing loss remained stable in these patients.

SNHL was irreversible in all cases patients.

In all patients with delayed FNP and in the patient with paralysis of an unknown onset there was complete resolution within 3-6 months. Only one patient with immediate paralysis experienced resolution with corticosteroid treatment after two months. Three patients needed surgery but the postoperative functional recovery was partial.

In patients with CSF leakage resolution occourred within 1-2 weeks in 7 cases (77.78%). Subtotal petrosectomy with obliteration by autologous fat was necessary for two patients with complete resolution after surgery.

- Most results are presented both as a table and a graph (Fig 2 and Table 1, Fig 4 and Table 2, Fig 5 and Table 3) or as Fig 3 and complete results text. Once is enough!

R: Thanks for this comment. We have deleted the graphs and improved the tables in order to make them clearer. Legends with full explanation of the tables have been included. We really appreciated your careful and thoughtful evaluation of our manuscript and hope that this revised version meets with your approval. Thanks again for your interest in our work. We await your review of our revised manuscript.

- The Discussion is very talkative and could be reduced in length. 

R: Dear Reviewer, thanks for careful review of our manuscript. The discussion was shortened in length.

Is the title of the study relevant? Nowhere in the text I can find any clear remarks that this is really in the expertise field of the otorhinolaryngologist.

R: Following your suggestions, we changed the title of the manuscript.

Round 2

Reviewer 3 Report

The revised manuscript has now been re-scrutinized. The authors have corresponded well to the suggestions made by the referee and also shortened the Discussion considerably.

There are however some difficulties concerning the English language and the manuscript could deserve being checked regarding language. I will give a few examples below on sentences changed or added in the revised manuscript:

  • Title: "Our experience on Temporal Bone Fractures: retrospective analysis on 141 cases"  - should be  "Our experience on Temporal Bone Fractures: retrospective analysis of 141 cases"
  • Page 2 "The aim of this study was to report our experience on TBFs treatment"  - should be  "The aim of this study was to report our experience on treatment of TBFs"
  • page 5, line 8  ".. CHL occourred.."  should be  "... CHL occurred.."
  • page 5, last section  "..all patients with delayed paralysis and in the patient with paralysis of an unknown onset, there was complete resolution within 3-6 months , in all cases."  should be "..all patients with delayed paralysis and in the patient with paralysis of an unknown onset, there was complete resolution within 3-6 months."

In summary I find this manuscript interesting and well reported. I recommend it for publication but prior to that it should be subjected to a screen regarding English language.